# Development of a Pandemic Resilience Competence Model for Healthcare Professionals—Individual and Organisational Aspects

**DOI:** 10.3390/ijerph22050712

**Published:** 2025-05-01

**Authors:** Nina Lorenzoni, Raquel Simões de Almeida, Daniela Wimmer, Ines Simbrig, Veronica Moscon, Fabio Carnelli, Nadine Sulkowski, Elohor Pamela Malaka, Paul Schober, Katharina Michel, Vítor J. Sá, Margit Raich

**Affiliations:** 1Department of Public Health, Health Services Research and Health Technology Assessment, UMIT TIROL—Private University of Health Sciences and Health Technology, Eduard-Wallnoefer-Zentrum 1, 6060 Hall in Tirol, Austria; daniela.wimmer@umit-tirol.at; 2LabRP-CIR, E2S, Polytechnic of Porto, Rua Dr. António Bernardino de Almeida, 400, 4200-072 Porto, Portugal; afa@ess.ipp.pt (R.S.d.A.);; 3Department of Nursing Science and Gerontology, UMIT TIROL—Private University of Health Sciences and Health Technology, Eduard-Wallnoefer-Zentrum 1, 6060 Hall in Tirol, Austria; ines.simbrig@umit-tirol.at; 4Institute for Public Management, Eurac Research, Viale Druso 1, 39100 Bolzano, Italy; veronica.moscon@eurac.edu; 5Center for Climate Change and Transformation, Eurac Research, Viale Druso 1, 39100 Bolzano, Italy; fabio.carnelli@eurac.edu; 6School of Business, Computing and Social Sciences, University of Gloucestershire The Park, Cheltenham GL50 2RH, UK; nsulkowski@glos.ac.uk (N.S.); pmalaka@glos.ac.uk (E.P.M.); 7Private Research Center, Hafelekar Consultancy, Society of Social Research and Education Eschenbachgasse 14, 3040 Neulengbach, Austria; paul.schober@hafelekar.at; 8DBU Digital Business University of Applied Sciences, Oranienstraße 185, 10999 Berlin, Germany; katharina.michel@dbuas.de; 9Centro ALGORITMI, School of Engineering, University of Minho, 4804-533 Guimarães, Portugal; 10Vice-Rectorate for Research and Development, University College of Teacher Education, Pastorstraße 10, 6020 Innsbruck, Austria; margit.raich@ph-tirol.ac.at

**Keywords:** resilience, competence model, healthcare, crisis, qualitative research, COVID-19

## Abstract

The COVID-19 pandemic highlighted the critical importance of resilience and adaptability at both individual and organisational levels in navigating unprecedented challenges. This study introduces a novel Pandemic Resilience Competence Model, a framework that articulates eight key competences each for individuals and organisations to enhance preparedness and response in pandemic scenarios. Employing a qualitative approach, the research identifies the essential skills and organisational capacities required to mitigate the impacts of pandemics. Using 50 semi-structured interviews with professionals and managers working in healthcare services in Austria, Germany, Italy, Portugal and the United Kingdom, the model provides actionable insights for implementing processes to improve preparedness and response in pandemic scenarios for stakeholders, including policymakers, educators, and organisational leaders. Findings highlight the interdependence of individual and organisational competences, reinforcing the need for integrated strategies to build pandemic resilience. The conclusions advocate for embedding the competences within training and development initiatives, aiming to enhance collective readiness for future global health crises.

## 1. Introduction

Crisis and pandemics significantly strain public health systems and healthcare professionals, intensifying pre-existing challenges such as high workloads, emotional demands, and resource shortages while introducing new pressures like exposure to an unknown virus, moral dilemmas, and prolonged uncertainty, all within volatile, uncertain, complex, and ambiguous (VUCA) conditions [1,2,3]. These circumstances push healthcare systems and organisations to their limits, requiring a proactive approach focused on preparedness, response, and recovery to maintain continuity of care [3].

Resilience is a vital competence for managing crises, sustaining operations, and ensuring patient safety [4,5]. Strengthening resilience involves fostering transparent communication, adaptive leadership, resource flexibility, and collaborative frameworks to enhance the capacity to anticipate, absorb, and recover from challenges [3].

Individual resilience refers to the ability of individuals to recover from adversity [6]. Organisational resilience, in turn, is the capacity of an organisation to absorb disturbances while maintaining essential functions and to adapt effectively to changing conditions during crises [7,8]. It also involves learning from past experiences and implementing proactive strategies to enhance long-term viability. These two forms of resilience are closely interconnected and together influence how well organisations perform during crises [9,10]. Building organisational resilience, therefore, requires a holistic, people-centred approach that supports adaptability and fosters a culture of continuous learning [10,11].

### 1.1. Individual and Organisational Resilience

The challenges of the COVID-19 pandemic have led to increased attention to the resilience of healthcare organisations. The healthcare system is characterised as complex, and its success is strongly dependent on a fluent integration and cooperation of multiple agents. Their willingness and ability to prepare, cope, and adapt are crucial to the overall functioning of the system [12]. Masten [13] (p. 101) defines resilience as “the capacity of a dynamic system to adapt successfully to disturbances that threaten system function, viability, or future development of the system”. Systemic resilience planning and acting aims to create a self-perpetuating dynamic that leverages and enhances the resilience capability of individual agents in a mutually supportive way, and thereby of the overall system. Healthcare organisations and professionals are components of the overall health system. Their interconnectedness with patients, communities, suppliers and health authorities must be understood and, thus, inform adaptations of resilience thinking for this sector [14]. For successful crisis management, special attention must be paid to the interplay of individual and organisational resilience, because strengthening the individual resilience will have a limited impact in coping with crisis, if resilience on the team and organisational level is not supported [15]. A robust level of individual and organisational resilience enables the healthcare system to respond effectively to major disruptions, maintain care and recover more quickly from the impact.

Individual resilience addresses the ability of a person to cope with intensive workload and stress during a crisis as well as to recover from such an event. It is considered as protective factor that helps individuals manage challenges and adverse events more effectively [16]. Trait-oriented resilience is seen as a stable personality trait that reflects the ability to return to a previous level of functioning after a stressful event. In contrast, process-oriented resilience refers to a dynamic process that involves not only recovery but also positive development beyond the original state. The latter approach emphasises proactive elements of individual resilience that can be trained and fostered [6]. 

To gain a comprehensive understanding of how an organisation can promote resili-ence, multiple interrelated factors and processes need to be considered [17]. Nemeth et al. [18] emphasise that resilience is about managing complexity and uncertainty and enabling organisations to navigate through crises effectively. Organisational resilience concerns the potential of an organisation to absorb disturbances while maintaining essential functions, to adapt effectively to changing circumstances and to learn from past experiences, thereby enhancing its long-term viability [19]. Beyond merely withstanding crises, it also encompasses the capacity to continue functioning and even thrive despite challenges [20].

Duchek [8] distinguishes three dimensions in her framework for organisational resilience: (a) Anticipation: this first phase is about recognising potential threats and critical developments at an early stage and being prepared for them. (b) Coping: in the second phase, the focus is on actively responding to crises or unexpected challenges. (c) Adaptation: the third phase emphasises the need for learning and continuous improvement following crisis situations. The aim here is to learn from past experiences, develop adaptive skills, and transform organisational knowledge in such a way that future resilience is strengthened. Resilience is also understood here as a dynamic process that requires continuous development and adaptation. It is, therefore, difficult to clearly delineate the three phases. Furthermore, these three dimensions are influenced by prior knowledge base, resource availability, social resources as well as power and responsibility.

Furthermore, Sutcliffe and Vogus [7] highlight the importance of a mindful organisation, which fosters a culture of awareness and adaptability, allowing for quick responses to unforeseen challenges. Jin et al. [21] complement this perspective by framing resilience as a dynamic process that involves continuous adaptation and transformation in response to crises. They highlight the critical role of leadership, culture, and communication in fostering resilience, suggesting that organisations must cultivate a supportive environment that encourages innovation and flexibility to navigate uncertainties and disruptions effectively. The questions that follow centre around what guidance is most appropriate and how organisations best embed this guidance into their strategic and operational thinking and practice.

### 1.2. Resilience in the Healthcare Sector

Resilience in the healthcare context can be defined as “the capacity to adapt to challenges and changes at different system levels, to maintain high quality care” [22] (p. 330). A set of capacities is needed to enable individuals and organisations to sustain high quality of care through adaptation, enhancement, and reorganisation [22].

Resilience-building efforts—whether at the individual or organisational level—directly impact employees’ ability to cope with these stressors, reducing risks of anxiety, depression, and burnout. Organisations that actively support mental health through professionals’ training, peer support and well-being systems, professional acknowledgment and leadership practices centred on empathy foster an environment where employees can thrive [23,24]. This emphasis on mental health is essential not only for employee well-being but also for maintaining workforce effectiveness and patient outcomes [25]. The intersection of resilience and mental health in the workplace is especially critical in healthcare settings, where employees are exposed to high stress and emotional demands [26].

Employee well-being is only one element required to maintain the effective functioning of healthcare organisations during crises. Czabanowska and Kuhlmann [27] stress the importance of developing the professional and, therefore, operational adaptation capability of individuals and teams. As has been argued elsewhere, this rests on crisis preparedness [28], devolved leadership [29], a shared understanding of issues and goals [30], the ability to maintain situational awareness, effective communication and learning capacity [31]. The process of building crisis resilience systemically, thus, requires the development of relevant competences at both individual and organisational level. It starts by identifying organisational crisis resilience capabilities and extends into understanding the role of organisations as ongoing and adaptive enablers of their workforce.

Furthermore, the development of health-specific standards (e.g., by International Organisation for Standardization (ISO) or the British Standards Institution (BSI) provide valuable guidance in ensuring that healthcare organisations are prepared to respond to pandemics and other crises. These standards not only promote consistent application of best practices but also strengthen the ability of systems to respond effectively to sudden changes or unexpected health threats. The BS65000 Organisational Resilience Standard [32] and the ISO22316 Security and Resilience [33]—Organisational Resilience Standards are the most holistic standards that integrate risk management, business continuity, and crisis management to ensure organisations can respond effectively to disruption. The standards provide detailed guidelines for developing a resilient organisation, including leadership, culture, and stakeholder engagement. Their focus on adaptability, agility, and innovation makes them useful for organisations seeking to enhance their crisis preparedness and response capabilities, and, thus, echo the thinking put forward by seminal authors in this field. However, these standards have limitations in tackling the specific challenges posed by pandemics or in addressing organisational pandemic resilience holistically.

The interplay between standardised protocols and adaptive flexibility enables continuous review and enhancement of health standards, promoting healthcare system resilience and post-crisis strengthening [17].

A successful combination of individual and organisational resilience in healthcare organisations during the pandemic was influenced by several interrelated factors. Effective leadership provides strategic direction and reduces uncertainty for the staff. Transparent communication supports a trustful environment and guarantees the alignment with goals during a pandemic [34]. The impact of organisational support systems, including emotional and psychological support by colleagues and managers, are critical in strengthening the individual resilience. These systems help healthcare workers to manage stress and maintain their well-being [9]. A further key element of resilient organisations is the adaptability of rapidly changing circumstances. This includes flexibility in terms of roles and processes [5] and creative problem solving that enables organisations to effectively overcome challenges of the pandemic [35]. Simões de Almeida et al. [24] also recommend healthcare organisations to offer continuous professional training to strengthen the individual resilience of their healthcare staff in equipping them with the necessary skills to strengthen their self-confidence and competence in successfully dealing with crises.

Finally, the integration of these elements fosters a resilient environment that supports both individual healthcare workers and the organisation. Thus, organisational resilience is not merely a reactive measure but a proactive strategy that integrates learning and adaptation into the organisation.

### 1.3. Research Gap

Resilience research is an emerging field that has gained momentum since the early 2010s, with key publications integrating psychological, social, and systemic perspectives. Kerr [36] established essential groundwork by developing competence-based frameworks that systematically embed resilience into organisational strategies, providing a structured approach to strengthening adaptability and preparedness.

The COVID-19 pandemic has accelerated interest in resilience, particularly within healthcare, leading to a surge in studies addressing individual, team, and organisational resilience. Recent works highlight critical factors influencing resilience, such as education, training programmes, and resource allocation [29,37]. However, the field still lacks sector-specific, empirically grounded studies offering holistic insights into organisational crisis resilience.

This study addresses the gap by presenting a novel competence model that integrates individual and organisational capabilities to enhance crisis resilience in healthcare settings. It builds on the increasing recognition of resilience as a core concept in disaster risk reduction, emphasising the need for systemic approaches, including disaster preparedness and access to emergency information, to mitigate the impact of large-scale health crises [38]. Thus, this study, part of the Erasmus+ project “Empowerment4Pandemias—Learning from COVID-19”, examines the challenges faced by healthcare professionals and organisations during the pandemic. It identifies effective coping strategies, adaptation measures, and solutions, culminating in the development of the Pandemic Resilience Competence Model. This model holistically conceptualises the competences required to enable healthcare organisations and their employees to better navigate crises, contributing to the broader goal of fostering resilience in the healthcare sector.

## 2. Materials and Methods

The collection of complex, context-specific experiences and perceptions by using qualitative approaches allows researchers to identify human behaviour and decision-making that are crucial in crisis situations [39]. For the sample selection, the researchers decided to include diverse groups of participants from different hierarchical levels, departments and organisations to gain insights that cover a wide range of perspectives and lead to more comprehensive results. Participants were recruited through existing contacts within healthcare institutions in the respective countries. Following initial outreach, a snowball sampling approach was employed, whereby interviewees recommended further eligible participants. This strategy was particularly suitable to access a diverse and experience-rich sample of healthcare professionals, some of whom may be considered hard to reach due to time constraints and workload pressures.

In total, 50 semi-structured interviews were conducted with professionals working in hospitals, nursing homes, or emergency services, with an equal distribution of 10 interviews per country across Austria, Germany, Italy, Portugal and the United Kingdom were conducted. Managers (care home directors, nursing directors, and medical directors in hospitals, people from health administration, nursing homes, and emergency services) and various professional groups with direct patient contact (doctors, nurses, therapists, paramedics) were interviewed. Table 1 gives an overview of the sample composition (for further details please see Appendix A).

The interview guides were developed in a collaborative and participatory process with nine project team members during a workshop. Based on prior joint literature research focusing on individual and organisational resilience in the crisis context of healthcare organisations, we jointly created a pool of questions, which were then grouped into thematic clusters for the final interview guide. A pre-test was carried out to ensure the validity and practicability of the questions.

Two interview guides (Appendix B), one to identify individual competences from healthcare professionals of all hierarchical levels and the other to determine organisational competences from the perspective of managers were developed. The interview questions addressed challenges during the pandemic, coping strategies, enabling factors, and the need for internal and external support.

The interviews in Austria, Germany, Italy, and Portugal were conducted between September and November 2022. The interviews in the United Kingdom took place later in May and June 2023 due to difficulties in accessing interview partners. However, the data obtained from the United Kingdom confirmed and complemented the patterns identified in the initial cross-country analysis. The interviews lasted between 30 and 75 min. Most of them were conducted in the native language of the interviewee, transcribed, and then translated into English by the project members. Depending on the country, between one and three project team members were involved in conducting the interviews. Some of the interviews were conducted face-to-face (n = 14), while others were conducted online via Zoom or MS Teams (n = 36).

The study was submitted to the Research Committee for Scientific and Ethical Questions (RCSEQ) of UMIT TIROL (N. º AZ3102). Prior to the interviews, all participants were given verbal and written information about the aims of the research project, the voluntary nature of their participation in the interviews, and were assured of anonymity. Those willing to participate signed an informed consent form.

### Data Analysis and Development of the Competence Model

To analyse the interview data, qualitative content analysis according to Kuckartz [40] was used. This method offers a structured approach for developing categories and systematically analysing the complexity and richness of interview data. This approach is particularly suitable for defining initial categories in a theory-driven way while remaining open to new topics that emerge from the data. MAXQDA24 software was used to support data organisation and the coding process.

After an initial structuring through deductive categorization based on the topics in the interview guide, an inductive processing of the texts took place in the second step, identifying topics emerging from the interviews. By ensuring the criteria quality and reliability of the data analysis, all interviews were coded by two researchers. Based on the results of this coding, the project team collaboratively developed the competence model. During a workshop, eight project team members systematically analysed and interconnected the presented findings. An iterative, collaborative approach was adopted, in which the team members contributed their respective perspectives and expertise to define the structure and central elements of the model. This process included critical reflection on the data, referencing existing literature and documented discussion of practical implications. Herberg and Torgersen’s [1] resilience competence model was considered, which, although not specifically designed for pandemics or the healthcare sector, explicitly accounts for the uncertainty factor that characterised the COVID-19 pandemic. Additionally, the WHO-ASPHER Competency Framework [41], which serves as a benchmark for public health education and workforce development, and the Comprehensive Hospital Agile Preparedness (CHAPs) model [28], which provides a guiding framework for hospital preparedness, addressing key areas of resource strain and offering adaptable strategies to enhance resilience in pandemic situations, were examined.

## 3. Results

The following section presents the identified competences for pandemic resilience on the individual and organisational level. The categories emerged from recurring themes in the interviews in how healthcare professionals described the challenges they faced, the strategies they used, and the conditions that supported or hindered their work during the pandemic. This conceptual framework aims to capture the experiences of the participants and translate them into actionable areas for fostering resilience in healthcare settings. Our analysis did not reveal country-specific or facility-specific differences in the lived experiences, challenges, or coping strategies described by participants. This may be attributed to the fact that healthcare professionals across different national contexts faced comparable challenges during the pandemic.

The model (Figure 1) is depicted as two circles within each other. The inner circle represents the eight competences for individual pandemic resilience, the outer circle the competences for organisational pandemic resilience.

### 3.1. Individual Pandemic Resilience Competences

#### 3.1.1. Emotional Awareness

Emotional awareness refers to the ability to recognise, express, understand, and process emotions. Accurately perceiving and regulating emotions enables healthcare workers to better cope with the emotional demands of their job, which can be especially important during a pandemic when they may experience increased stress, anxiety, and fear. Emotional awareness helps to identify and manage stress, which is important to prevent burn-out and maintain an overall mental well-being in high demanding situations like pandemics. High emotional awareness enhances understanding of how emotions impact individual perception and decision-making. Furthermore, it helps in better recognising the needs of colleagues and patients, thereby enabling an effective and compassionate response during a pandemic or in a crisis situation. One interviewed nurse captured this connection between self-awareness and care for others by stating “And I think it was also good to look after yourself and to do something good for yourself. Because you are there for others a lot in the medical profession, so you still look after yourself and look for little moments of happiness and treat yourself well too”.

#### 3.1.2. Role Awareness

Role awareness means maintaining awareness of one’s own competences and scope of practice while also recognising the need for adaptability during a pandemic. In crisis situations, professional and social roles may change rapidly, requiring individuals to quickly understand, learn, and adjust to new professional and social responsibilities within new or evolving team structures. One interviewee reflected “These challenging moments prompted me to brief young colleagues in critical situations, if possible, before entering the situation, and to actively offer myself as a peer to discuss the situation and offer possible solutions and support”. Role awareness includes adherence to duties, obligations, and codes of conduct defined by occupational standards, legal regulations, and organisational procedures, including any updated protocols and guidelines related to the pandemic. When encountering situations beyond one’s competence or scope of practice, guidance is sought. Ensuring that individuals understand and adapt to their roles facilitates coordination, which is crucial in crisis situations where resources are limited, decisions must be made quickly, and effective teamwork is essential.

#### 3.1.3. Ability to Communicate 

The ability to communicate influences both how healthcare workers share their knowledge among each other and how they guide, inform, support and collaborate with their patients and colleagues. “In hindsight, the most important thing was probably to create a sustainable structure first, to establish good communication, especially with patients, but also with all external partners.”, concluded one interview participant. Another interviewee described: “Many times I would wait for the night shift to arrive to explain to them why certain measures had to be changed and how it was to be done and the importance of doing it, and I wanted to be the one to pass the message on so that there would be no gaps in the information here, so that everyone would be in agreement”.

Healthcare workers need to be proactive in managing these interactions, considering the heightened level of anxiety and stress that many individuals experience during a pandemic. This requires adapting communication to the goals, needs, urgency, target group, and sensitivity of each interaction. In addition, conveying information in a purposeful and clear manner is crucial, especially in the context of a pandemic, where there is a high demand for accurate and up-to-date information. This involves effectively managing the dissemination of information and documentation to ensure that all stakeholders have access to the information they need. Effective communication entails more than the mere exchange of information; it encompasses essential emotional and psychosocial dimensions.

#### 3.1.4. Creativity and Improvisational Skills

Creativity and improvisational skills refer to the ability to use available resources and knowledge in new and innovative ways to address the challenges of providing best possible care during a pandemic. Healthcare workers may need to develop new and innovative strategies to provide care, protect themselves and their patients. One of the medical doctors described “What was absolutely crucial for me was the courage—or perhaps the desperation—to abandon all standard procedures and improvise”. Creative thinking can lead to the development of new medical interventions, technologies, and protocols that can help to mitigate the impact of the pandemic on patients and healthcare workers alike. Additionally, improvisational skills are essential in adapting healthcare services to rapidly changing circumstances during a pandemic. 

#### 3.1.5. Ability to Maintain Sound Relationships

The ability to maintain sound relationships refers to the capacity to foster positive relationships based on psychological safety, well-being, openness, trust, and support. Especially during a pandemic, when healthcare workers face increased stress and pressure, it is important to prioritise good relationships and team cohesion. One interviewee emphasised the key role of direct supervisors in this regard: “The ward managers are the ones who have the most influence, who give structure and security. Because it’s the ones who stand there, the ones who are tangible, who comfort people, build them up, motivate them, give them support”. Creating a sense of community and support is essential to maintain well-being and cope with the demands of the job. This includes maintaining open and transparent communication, showing empathy and sharing emotions. The ability to maintain sound relations is essential in promoting trust, confidence, and cooperation, which can help to mitigate the mental impact of pandemics. Beyond the workplace, seeking support from friends and family can also be beneficial for mental health. Most interviewees described their families and friends as most important support network during the challenging times of the pandemic.

#### 3.1.6. Situational Awareness and Preparedness 

Situational awareness is the ability to perceive, comprehend, and anticipate the situation. It includes understanding the potential risks and being able to make informed decisions based on this. The healthcare setting is complex and constantly changing, and the circumstances of a pandemic further exacerbate this dynamic. Recognising and comprehending the situation is critical to anticipate changes and deliver best possible care under pandemic circumstances. Pandemic preparedness refers to measures taken to prepare for and reduce the effects of pandemics. This includes predicting and preventing the spread of pandemics, mitigating their impact on vulnerable populations, and effectively managing their consequences. It involves developing pandemic plans, training healthcare personnel, educating communities, and regularly monitoring and evaluating preparedness measures. The feeling of being well prepared and being able to assess the situation was described as helpful: “I think it was good to have prior experience that you had already dealt with illnesses that you have to isolate like MRSA or something like that, so you could already do that and also the protective measures and the targeted training that you have every year in advance were beneficial”.

#### 3.1.7. Professional Competence

Professional competence helps to remain calm and focused, also in the face of a pandemic. Being confident in one’s professional abilities and having the skills and knowledge to handle difficult situations, helps to manage the challenges that arise during a crisis like a pandemic. One interviewee summarised “After 25 years in service you simply have a portfolio that you can also draw on in crisis situations”.

#### 3.1.8. Ability to Reflect and Learn

The ability to reflect and learn refers to a continuous process of simultaneous real-time adaption and systematic post crisis evaluation. Taking time to ponder what happened, and critically assessing what went well, and what could have been performed differently is essential not only in the aftermath of a pandemic but also throughout its course. Based on these adaptations and improvements for preparedness and response to future pandemics can be made. To effectively reflect and learn, one must create room for having a reflexive mindset, taking a step back, and ask and discuss critical and relevant questions. Through capturing and utilising all the elements from past experiences, one can continually improve resilience, and the services delivered in the next pandemic or crisis. However, finding the time on a regular basis becomes often challenging, as one interviewee observed, “I think the management staff would have needed training as well. It seems to me that when you’re young, you go through a lot of training. And then at some point, you’re in a leadership position and no longer receive regular training—especially not in how to handle a crisis situation”. Another participant highlighted the challenge of sustaining reflective practices in the face of operational pressures: “It would be important for all those involved in pandemic management to sit down together and work through what happened, but then you know, it always gets lost in the day-to-day business”.

### 3.2. Organisational Pandemic Resilience Competences 

#### 3.2.1. Risk and Threat Understanding

Risk and threat understanding involves culture, processes and structures that are established to understand and effectively manage pandemics in the primary healthcare sector and to reduce risks to patients, staff, suppliers, the wider public, and the organisation to an acceptable level. This includes the ability to develop ongoing situational awareness and to conduct appropriate cost–benefit analyses of risk reduction strategies. As one nurse reflected, “There were areas, though, where I felt that things could have been handled differently. For example, we would wear PPE at work, but then we would still commute home on crowded buses. Additionally, if we cared for a COVID-19 patient, we would then go back to caring for other patients who didn’t have COVID-19. We didn’t have enough staff to isolate those who had been exposed to the virus”. This highlights how gaps in systemic planning and resource allocation can undermine risk management strategies, emphasising the need for integrated approaches.

#### 3.2.2. Strategic and Operational Thinking

Strategic and operational thinking encompasses a full understanding of how the organisation is running and governing during a pandemic. This includes preparedness and operational adaptations, in line with relevant health protocols, to meet the needs of patients and to protect healthcare workers, suppliers and the wider public. As an example, one interviewee explained “During the pandemic, most organisations were short of staff […] because so many people were out of work. But I don’t think we had that in my place of work. Whenever anybody called in sick with COVID or any other medical condition, there was always provisions for a replacement”. It requires demonstrable evidence that the organisation is not complacent, fully compliant and proactive in minimising the spectrum of risks that arise from a pandemic.

#### 3.2.3. Leadership for Resilience and Integrity

Leadership for resilience and integrity focuses on responding quickly, appropriately and in line with public health protocols to the pandemic threat, demonstrating competent decision-making and accountability across organisational structures. Implementing a response based upon a culture of trust, transparency, empathy and innovation capability that allows the organisation to continue functioning efficiently and to meet its obligations towards its patients, staff, suppliers and the wider public. One leader described his personal approach to being visibly present and accessible during the early phase of the pandemic: “So in the first wave in particular, where the issue of fear and uncertainty was very strong, I was in the building every day for 46 or 47 days in a row. I went through all the wards every day and it was very, very important to me that I was close to the staff and accessible to them”.

At the same time, other participants expressed a desire for more supportive and empathetic leadership under extreme conditions. One interviewee reflected “I would have liked a little more understanding. We’ve endured things that you can’t actually endure”.

#### 3.2.4. Information Management

Information management refers to the ability to continue to manage information—physical, digital and intellectual property—according to the standard protocol, whilst ensuring the timely and authoritative sharing of pandemic related information. This includes being trusted to safeguard sensitive information whilst meeting necessary reporting requirements. This requires the adoption of information security—minded and dynamic practices that allow stakeholders to gather, store, access, share and use authoritative information securely, effectively and in a timely manner.

The interviewees described different ways in which fast and reliable information management was handled in their organisation. E-mails, employee apps, notice boards, intranet, and face-to-face exchanges at handover meetings were frequently cited examples. Provision via different channels was seen as particularly important: “The COVID Taskforce meetings resulted in a protocol that was forwarded. And then we repeatedly summarised so-called Infonews in paper form and sent it to all employees, via the mailboxes and also via the internet, so that there were various sources for passing on information”.

#### 3.2.5. Adaptability and Flexibility

This competence implies the ability of organisations to successfully handle changing circumstances, that may range from minor to disruptive eventualities, whilst minimising, as much as possible, adverse impacts on core operations. This means that managers must also be able to adapt to the challenges of a crisis. However, this is not always so easy, as one interviewee described “At that moment, it was possible to determine quite quickly who was suitable for crisis management and who was not. [...] Crisis means that you no longer have control or have limited control. And there are people who can deal with it better. And there are people who are less good at it”.

#### 3.2.6. Focus on People and Culture 

A focus on people and cultures includes how the organisation supports staff and ensures individual preparedness while maintaining safe work environments. One interviewee, who worked in a nursing home, summarised it “A good response to a pandemic from an organisation is […] prioritizing the welfare of both the people we support and staff members at the time of the pandemic”. This also includes how healthcare workers interact with patients and their families, and how the organisation is perceived to interact with relevant authorities, its supply chain partners, and the wider public during the pandemic. Some interviewees were supported by their employers were supported by their employer with temporary accommodation to prevent possible infection of their families. Consequently, the organisation understands that it will be judged by the personal experience individuals have with it during the pandemic. Yet several participants described feeling a lack of support, particularly for staff well-being. As one stated “If we go to our boss and say we need supervision, we always get no for an answer”.

#### 3.2.7. Training, Testing, and Validation

This competence implies the examination of a pandemic plan that addresses multiple components, in conjunction with each other, typically under simulated operating conditions. This involves the following: (i) an analytical methodology which considers concurrent and contextual review of multiple metrics, to provide a more complete picture regarding the pandemic plan; (ii) regular exercises and testing conducted on multiple interrelated components of the pandemic plan, typically under simulated operating conditions to ensure workability and role understanding across the workforce; (iii) testing and training should extend beyond regular plan assessments to include the testing and training of other institutional resilience segments. One nurse described the importance and the resulting confidence of regular trainings “The most important aspect during the pandemic was infection control. I had previous training in infection control […] and it was mandatory for everyone to stay up-to-date. This knowledge proved invaluable during the pandemic because I knew the appropriate measures to take for every patient. […] Having received prior training, applying it during the pandemic, and benefiting from ongoing support by the infection control team all played a crucial role in my ability to handle the situation”.

#### 3.2.8. Ability to Reflect and Learn

The ability to reflect and learn in an organisational context is based on four core capabilities: (i) Adaptive Capacity: reflecting the ability to react to emerging, immediate and sustained pandemic situations; (ii) Horizon Scanning: the ability to examine information to identify approaching changes, including threats and opportunities; (iii) Innovation: an organisation’s ability to be innovative during a pandemic and on retrospection; and (iv) Learning Capacity: drawing and learning from conclusions for forward planning However, participants described major obstacles to fully realising these capabilities in practice. As one interviewee put it, “What I should actually be doing is dealing with the aftermath of the last crisis, but we don’t have the time. Neither we in management nor the employees at the grassroots level. And that leads to this accumulated stress”. Another emphasised the importance of institutionalising learning for the future: “And if possible, take the experiences you have had with you, write them down and take the time for pandemic plans too—who knows, maybe in 5–10 years you will need them again”. These reflections show how structural constraints, such as time pressure and staff shortages, can hinder both immediate and long-term learning processes within healthcare organisations.

A cross-cutting theme that emerged across all interviews was the persistent and severe shortage of staff in the healthcare sector. This structural issue not only shaped the conditions under which the pandemic was managed but also poses an ongoing challenge to the development and implementation of the resilience competences identified in this study. In contexts where time and human resources are already stretched to the limit, fostering reflection, collaboration, and adaptation becomes significantly more difficult. As one interviewee, a nursing director, put it “If there is enough staff, resilience is more likely to be fostered than if staff are pushed to their physical and psychological limits in every service. The pandemic has once again shown where the strengths and weaknesses are in the health system, and ultimately we need the people who work in it—and they have to be found and trained”. A third interviewee noted the frustration experienced on the ground: “Yes, we would have liked to have additional staff to relieve the burden, but this was rejected”. Overall, the results indicate that resilience is an ongoing process that needs to be fostered in times of stability outside of the crisis. One of the medical doctors interviewed summarised this: “Therefore, we must continue to convey the message that people were able to respond to a critical moment, serious isn’t it, of ignorance, but after the crisis they continue to be extremely important for us to be able to continue to function on a daily basis. Resilience is daily, it is not resilience in a crisis. This is what is important to pass on to people, I think”.

## 4. Discussion

This study aimed to develop a comprehensive competence model to enhance pandemic resilience in healthcare settings, addressing both individual and organisational aspects. The findings, derived from qualitative interviews conducted across five countries, offer valuable insights into the lived experiences of healthcare professionals and managers and their strategies for navigating the challenges of the COVID-19 pandemic. The competence model emphasises the interplay between individual and organisational competences, recognising their mutual reinforcement as central to building holistic resilience in healthcare systems.

Competence models are systematic representations of relevant competences [42]. They can be used, among other things, for designing learning processes and programmes. The identification and documentation of relevant action competences is fundamental in the development of curricula for educational programmes, particularly in vocational training, as it helps promote an orientation toward practical actions and enables learning that is closely connected to real-life, professional, or organisational contexts. From the identified competences to be acquired, learning objectives, outcomes, and consequently, learning content are derived. Moreover, competence models serve as a guide for didactic approaches and methods in teaching. On the other hand, learners can better understand which skills and abilities are necessary for successfully mastering, for example, a specific professional field [41,43].

From a systems perspective, competence models can be designed for different components or spheres of the system and can be aimed towards developing systems-relevant competences at, for example, the individual, organisational, or institutional/governance level. To ensure overall cohesion, competence models designed for individual system components must, therefore, build in interdependencies and the creation of synergies, as pointed out in the WHO-ASPHER Competency Framework, which serve as the point of departure for a variety of different activities aiming to strengthen the public health workforce in the European Region [41].

Hence, our proposed competence model intends to be a framework that outlines the specific skills, knowledge, and behaviours that are necessary for success in a healthcare job or healthcare organisation. Our model highlights eight critical competences for individual resilience, including emotional awareness, creativity, adaptability, and reflective practice. These competences enable healthcare professionals to manage stress, respond to rapidly changing circumstances, and maintain their psychological well-being during crises [10,17]. Complementing this, organisational resilience is underpinned by eight competences such as leadership for resilience, resource flexibility, and strategic risk management, which collectively foster an adaptive and supportive work environment [4,44].

The dual-circle structure of the model emphasises the interconnectedness of individual and organisational resilience. By emphasising the need for an integrative approach, the model aligns with prior research advocating for systemic strategies that address the cultural, structural, and operational dimensions of healthcare organisations [45,46]. This integrative perspective is especially crucial in complex systems like healthcare, where diverse stakeholders and interdependencies influence outcomes [47]. Duchek [8] also emphasises the interactions between the dimensions of organisational resilience and underlines the importance of a supportive organisational culture that promotes employee resilience. Building a resilient organisational culture can promote individual resilience. For example, if an organisation values training, support and flexibility, employees can respond better to change and develop their individual resilience skills. The availability of resources, including social resources and a sense of community, which operate at both an individual and organisational level, plays an important role in the development of resilience. People who work in a supportive and resilient organisation often have a stronger emotional and social base that strengthens their individual resilience. According to Lyng et al. [17] leadership plays a particularly important role because good leadership can lay the foundation for other important factors in promoting resilience.

In their study on the resilience of healthcare managers, Förster et al. [11] also point out the interaction between individual resilience factors. Situational factors, such as the work environment, organisational support, and social networks interact with individual experiences to increase resilience. A supportive environment can facilitate learning from past experiences, while negative or stressful situations can strain individual resilience. However, organisational practices intended to enhance resilience may inadvertently lead to exploitative situations. The expectation for individuals to be resilient can create a situation where employees feel an undue burden to cope with systemic challenges without adequate organisational support. This can lead to increased mental health issues, burnout, and a sense of personal failure if individuals struggle to meet those resilience expectations while lacking institutional support [48].

The cross-national nature of our study showed that, despite differences in health systems and organisational structures, healthcare professionals relied on similar coping strategies with the demands of the pandemic.

These findings highlight the relevance for health systems to invest in the long-term development of resilience competences, at both individual and organisational levels. Beyond the immediate response to the pandemic, the study findings point out the need of integrating resilience strategies into everyday healthcare practice and policy. The findings reinforce the importance of strengthening resilience during periods of stability rather than waiting for crises to arise. Recent evidence shows that proactive resilience-building, through targeted initiatives such as simulation exercises and continuous professional development, fosters a culture of preparedness that mitigates the impact of future disruptions [3]. Resilience should be cultivated as a continuous process, incorporating structured training, strategic resource allocation, and systemic improvements that reinforce adaptive capacity at both the individual and organisational levels. In healthcare, where the stakes are exceptionally high, fostering resilience transcends strategic considerations; it represents an ethical commitment to safeguarding the well-being of both employees and patients. Moreover, proactive resilience fosters a positive outlook and problem-solving mindset, enabling organisations to not only withstand crises but also to emerge stronger. Therefore, shifting from a reactive to a proactive resilience framework is essential for healthcare organisations aiming to sustain long-term success in an unpredictable environment [49].

Additionally, developing a resilient workforce requires investments in training programmes, flexible leadership, and a culture of empathy and collaboration. Healthcare organisations must prioritise initiatives such as transparent communication, role clarity, and supportive management practices to mitigate psychosocial risks [50]. As stated by the World Health Organisation and the International Labour Organisation [51], while decent work positively impacts mental health by fostering purpose, structure, and relationships, poor working conditions pose significant risks. Psychosocial risks at work, such as high demands, low job control, unclear roles, and unsafe environments, contribute to stress, burnout, anxiety, depression, and even suicidal behaviours. Factors like discrimination (based on race, gender, disability, etc.), financial insecurity, unstable employment, and inequality in opportunities exacerbate mental health issues. These risks are particularly relevant in healthcare, where demanding schedules and high-pressure environments can compound stress and mental health vulnerabilities. Improving mental health at work is crucial for upholding the human right to good health and progressing towards the Sustainable Development Goals (SDG), particularly SDG 3 on health and SDG 8 on decent work for all. The WHO [52] estimates that common mental disorders like depression and anxiety result in a global economic loss of 1 trillion USD annually due to decreased productivity. As workplace mental health is closely linked to individual and organisational resilience, employees who receive adequate mental health support demonstrate higher adaptability and sustained productivity, even in the face of crises. Therefore, organisations that implement structured mental health policies, not only enhance employee well-being but also strengthen their capacity to navigate uncertainties.

It is possible to conclude, that the model’s competences can inform organisational development, human resource strategies, and educational curricula for healthcare professionals. For example, incorporating resilience training into daily practices and leveraging digital tools like virtual reality simulations could enhance individual and team preparedness for future crises [53]. Furthermore, the adaptability of the model to non-healthcare sectors such as education and emergency response offers opportunities for broader application and cross-sectoral learning.

### 4.1. Strengths and Limitations

The strengths of this study include its qualitative methodology, multi-country validation process, and inclusion of diverse healthcare professions and management levels, which ensure a multidimensional perspective. However, limitations such as potential recall bias, social desirability in interview responses, and reliance on qualitative data should be acknowledged. While such experiential accounts reflect subjective perspectives and sense-making processes rather than objective or generalisable truths, they nonetheless offer valuable insight into organisational reality and provide a meaningful basis for conceptualising key areas of resilience in healthcare settings. Future studies could complement these findings with quantitative analyses to measure the model’s impact on outcomes such as employee well-being, organisational continuity, and patient safety.

### 4.2. Future Research Directions

Building on the current findings, future research should explore deeper into the interdependencies between individual and organisational competences, as highlighted by Lyng et al. [17]. Longitudinal studies could assess the long-term effects of implementing the competence model on healthcare resilience and crisis preparedness. In addition, investigating the model’s adaptability to diverse cultural and healthcare system contexts, particularly outside Europe, would enhance its global relevance.

Emerging technologies offer promising avenues for advancing resilience training. For instance, artificial intelligence-based platforms and virtual reality tools could be used to simulate crisis scenarios and strengthen specific competences such as decision-making under pressure and adaptive leadership [54]. Finally, research on policy integration could facilitate the inclusion of resilience frameworks in national disaster preparedness plans, contributing to sustainable healthcare systems capable of withstanding future crises.

## 5. Conclusions

The presented competence model for resilience serves as a foundational framework for strengthening both the coping skills of healthcare professionals and the structural stability of healthcare organisations. In high-stress environments such as healthcare, where mental strain and systemic challenges are pervasive, fostering resilience-promoting competences is essential to ensuring long-term individual well-being and organisational sustainability. By addressing the complex interplay of individual and organisational competences required for effective crisis management, this model provides a comprehensive approach to navigating the challenges posed by pandemics and other systemic disruptions. Beyond its theoretical contributions, the model offers approaches for developing training programmes, advancing organisational development, and informing policy design aimed at creating resilient healthcare systems.

While this study highlights the model’s relevance across European healthcare systems, further research is required to assess its adaptability in diverse contexts, quantify the impact of targeted interventions, and explore the interdependencies between competences. Future studies should also investigate how resilience can be proactively integrated into everyday practices to better prepare healthcare systems for future crises.

The competence model demonstrates the ethical and strategic importance of building resilience in healthcare as a means of safeguarding professionals’ mental health, enhancing organisational preparedness, and ensuring the continuity of high-quality care. As the frequency and complexity of crises increase, adopting integrative and context-sensitive resilience strategies becomes imperative for sustainable healthcare systems worldwide.

## Figures and Tables

**Figure 1 ijerph-22-00712-f001:**
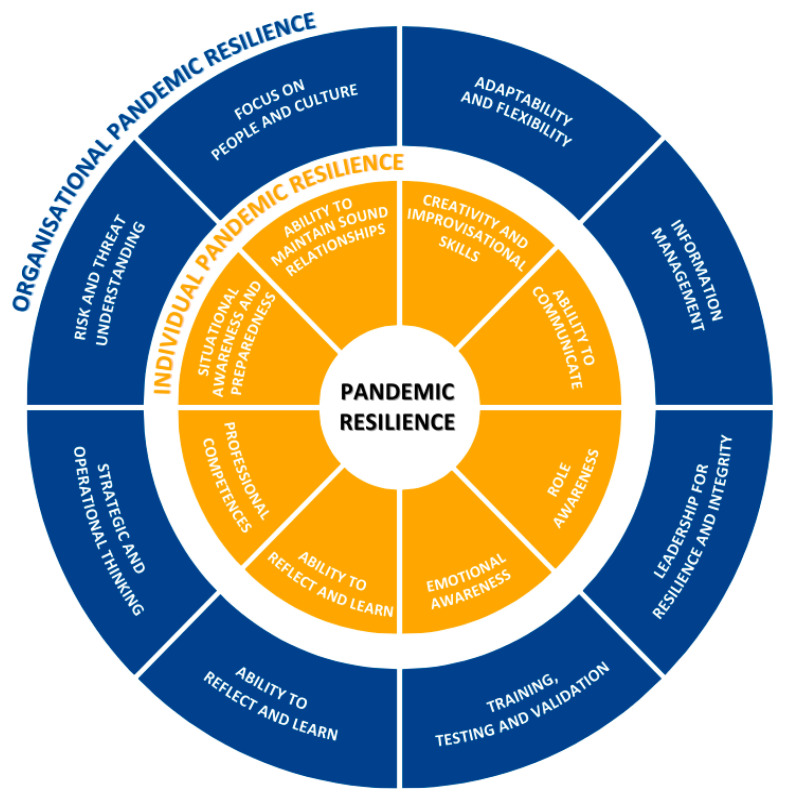
Competence Model for Pandemic Resilience.

**Table 1 ijerph-22-00712-t001:** Characteristics of Interview Participants.

Country	Participants	Gender F/M	Age Range	Occupations
Austria	10	5/5	25–73	Medical Doctors, Nurses, Physiotherapist, Nursing Director, Dietician, Emergency Doctor, Nursing Home Director;
Germany	10	6/3 *	31–58 **	Physiotherapists, Nurses, Nursing Director, Health Administration Workers, Health Technology Specialist;
Italy	10	4/6	38–64	Nurses, Medical Doctors, Nursing Directors, Health Administration Workers, Social Worker, Lawyer at an emergency service;
Portugal	10	8/2	29–62	Occupational Therapists, Physiotherapist, Nurses, Medical Doctors, Radiology Technician, Member of Hospital Board;
UK	10	6/4	n/a	Nurses, (Senior) Healthcare Assistants, Operations Support Manager, Team Leader Care Home;
Total	50	29/20 *	25–73 ***	

* One participant did not indicate gender. ** Age range excludes 6 participants with missing age information. *** Age range based on 39 participants with available age data.

## Data Availability

The data that support the findings of this study are available from the first author (NL) upon reasonable request.

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
