# Peer review of "Development of a Pandemic Resilience Competence Model for Healthcare Professionals—Individual and Organisational Aspects"

_ijerph, 2025, doi:10.3390/ijerph22050712_

Round 1
Reviewer 1 Report
Comments and Suggestions for Authors
The manuscript addresses a highly relevant topic which is resilience in healthcare. The paper has a fine structure, and it is possible to observe an important line of coherence across all its main steps. The paper is well written and relates to important literature in the field. However, I highlight areas of improvement that need to be addressed before the manuscript can be accepted for publication:
- My main comment if related to the use of qualitative data. Data emerging from qualitative interviews can bring an important contribution by exploring how people experience and make sense of a certain phenomenon. However, although sense-making and experiential accounts of participants represent an important dimension of organizational reality, these should not be presented as “hard fact” evidence of how things are. The presentation of finding on section 3 has formulations which may indicate relations between variables that can not be assumed by only looking at qualitative data. It is important to make it clear that the data describes experience of the participants. This is in itself a rich exploration that can inform a resilience model as suggested by the authors.
I would recommend a thorough revision of the presentation highlighting the qualitative character of the data presented and making the voice of the participants more vivid by incorporating quotes from interviews that the authors may regard as particularly representative of the findings being described.
- I would recommend adding the interview guides as appendixes, rather supplementary material. The methods part needs to describe in more detail how the interview guide was composed. Does it operationalizes theories/concept from prior research? How?
- The paper analyzes data from 5 countries. The study does not follow a comparative character, but it could still be enriching to discuss differences in findings across these countries. Were there differences?
- One of the most important resilience models presented in recent years is the one by Duchek, S. (2020). Organizational resilience: a capability-based conceptualization. Business research, 13(1), 215-246 and later applied by many others such as Shaya, N., Abukhait, R., Madani, R., & Khattak, M. N. (2023). Organizational resilience of higher education institutions: An empirical study during Covid-19 pandemic. Higher education policy, 36(3), 529-555 and Bento, F., Garotti, L., & Mercado, M. P. (2021). Organizational resilience in the oil and gas industry: A scoping review. Safety science, 133, 105036.
Resilience is presented here as process and a capability. I would recommend briefly presenting this model on the literature review part of the manuscript and then discussing on section 4 how the model suggested by this manuscript could be related (differences? similarities? complementarity?) to Duchek’s model. This would contribute to highlight the theoretical contribution of this manuscripts.
I wish the authors all the best in revising and resubmitting this interesting paper.
Author Response
Thank you very much for your valuable feedback. Please see attachment for our detailed response.

Reviewer 2 Report
Comments and Suggestions for Authors
This manuscript describes the Pandemic Resilience Competence Model, by highlighting competencies needed by healthcare professionals at the individual and the organizational level. Overall, the paper is well-written, and the authors identify and describe competencies at each level. A major comment is the need for qualitative data to support the findings, which is missing from the results. Attached are specific comments. I hope you will find them to be constructive and helpful.

Author Response
Thank you for your valuable feedback. Please see attachment for our detailed response.

Round 2
Reviewer 1 Report
Comments and Suggestions for Authors
The authors have addressed all my comments in their revision of the article and I therefore recommend it for publication.
Author Response
We thank the reviewer again for the valuable feedback, which helped to substantially improve our manuscript and now the recommendation for publication.
Reviewer 2 Report
Comments and Suggestions for Authors
Congratulations in the significant improvements to the paper. However, there are a few minor revisions needed.
- The introduction should be reviewed again to ensure that definitions of resilience and other factual statements are appropriately cited.
- The results section still needs more work.
- Although the demographic information is included in the supplementary file, consider providing a table in the main document that summarizes the characteristics of the participants by county, gender, age range and occupation. Thank you for including the participant quotes.
- Please include the general characteristics of the individuals associated with each quote (Occupation, age and Country).
- Proofread for additional grammatical errors.
Author Response
Thank you for your feedback, please see attachment for our detailed response.
